# RNA polymerase II-mediated rDNA transcription mediates rDNA copy number expansion in *Drosophila*

**George J. Watase** [1,2]*, **Yukiko M. Yamashita** [2,3,4]*

**1** Department of Germline Development, Institute of Molecular Embryology and Genetics, Kumamoto University, Chuo-ku, Kumamoto-shi, Kumamoto, JAPAN, **2** Whitehead Institute for Biomedical Research, Cambridge, Massachusetts, United States of America, **3** Massachusetts Institute of Technology, Department of Biology, Cambridge, Massachusetts, United States of America, **4** Howard Hughes Medical Institute, Cambridge, Massachusetts, United States of America

* gwatase@kumamoto-u.ac.jp (GJW); yukikomy@wi.mit.edu (YMY)

**Data Availability Statement:** All relevant data are within the paper and its Supporting Information files.

**Funding:** Howard Hughes Medical Institute (to Y.M.Y), the John Templeton Foundation (to Y.M.Y,

## Abstract

Ribosomal DNA (rDNA), which encodes ribosomal RNA, is an essential but unstable genomic element due to its tandemly repeated nature. rDNA's repetitive nature causes spontaneous intrachromatid recombination, leading to copy number (CN) reduction, which must be counteracted by a mechanism that recovers CN to sustain cells' viability. Akin to telomere maintenance, rDNA maintenance is particularly important in cell types that proliferate for an extended time period, most notably in the germline that passes the genome through generations. In *Drosophila*, the process of rDNA CN recovery, known as 'rDNA magnification', has been studied extensively. rDNA magnification is mediated by unequal sister chromatid exchange (USCE), which generates a sister chromatid that gains the rDNA CN by stealing copies from its sister. However, much remains elusive regarding how germ cells sense rDNA CN to decide when to initiate magnification, and how germ cells balance between the need to generate DNA double-strand breaks (DSBs) to trigger USCE vs. avoiding harmful DSBs. Recently, we identified an rDNA-binding Zinc-finger protein Indra as a factor required for rDNA magnification, however, the underlying mechanism of action remains unknown. Here we show that Indra is a negative regulator of rDNA magnification, balancing the need of rDNA magnification and repression of dangerous DSBs. Mechanistically, we show that Indra is a repressor of RNA polymerase II (Pol II)-dependent transcription of rDNA: Under low rDNA CN conditions, Indra protein amount is downregulated, leading to Pol II-mediated transcription of rDNA. This results in the expression of rDNA-specific retrotransposon, R2, which we have shown to facilitate rDNA magnification via generation of DBSs at rDNA. We propose that differential use of Pol I and Pol II plays a critical role in regulating rDNA CN expansion only when it is necessary.

Grant #: 62456), and PREST, JST (to G.J.W, Grant #: JPMJPR2389). The funders had no role in study design, data collection and analysis, decision to publish, or preparation of the manuscript. Yukiko Yamashita received salary from Howard Hughes Medical Institute. George Watase's salary was funded by research budget from Howard Hughes Medical Institute provided to Yukiko Yamashita for a part of time this research was conducted.

**Competing interests:** The authors have declared that no competing interests exist.

## Author summary

Ribosomal DNA (rDNA) exists as tandemly-repeated copies in eukaryotic genome, making it unstable due to spontaneous intrachromatid recombination that causes copy number loss. The germline, which is the sole cell type that transmits genome from generation to generation, must expand rDNA copy number to counteract spontaneous copy number loss. It has been shown that the process of rDNA copy number expansion, called rDNA magnification, involves rDNA binding protein Indra and retrotransposon R2. However, the underlying molecular mechanism of rDNA magnification and how Indra and R2 may functionally intersect remained unknown. This study shows that Indra is a transcriptional repressor of the intergenic spacer (IGS) sequence of rDNA, orchestrating R2 expression and rDNA copy number expression. Low rDNA copy number downregulates Indra, leading to expression of IGS, which in turn leads to expression of R2. It was found that IGS expression is mediated by RNA polymerase II, which is specifically recruited to nucleolus upon rDNA copy number reduction. Together, the results lead to a model how rDNA copy number reduction triggers the process of rDNA copy number expansion.

## Introduction

Metazoan's germline has continued through more than a billion years of its evolutionary history. The germline faces the challenge of maintaining genome integrity, particularly genomic elements that tend to erode over time, such as telomeres and ribosomal DNA (rDNA) [1–3]. rDNA, which encodes for ribosomal RNA (rRNA), is a highly repetitive genomic element in the eukaryotic genome. While its repetitiveness is critical to meet the high demand of ribosome biogenesis [4], its repetitiveness causes spontaneous intrachromatid recombination at rDNA loci, leading to copy number (CN) reduction while generating extra chromosomal rDNA circle (ERC) (Fig 1A) [5]. rDNA CN reduction on the chromosome as well as accumulation of ERCs are proposed major causes of replicative senescence in yeast [6,7], although the causal link is yet to be established [8]. rDNA CN reduction has also been observed during aging in a wide range of multicellular organisms [9–11], thus the germline must counteract such CN loss to sustain its 'immortality', the ability to pass the genome perpetually from generation to generation.

Extensive studies have established the framework of how yeast cells eliminate ERCs and recover chromosomal rDNA CN to reset their replicative age: homologous recombination-mediated process is known to increase rDNA copy number [12]. In parallel, ERCs associate with the nuclear pore complex within the mother cells, leading to their retention in the mother cells, which causes the aging of the mother cell while rejuvenating the daughter cell [13,14]. Studies in the yeast established that rDNA copy number maintenance is a key to lineage survival.

*Drosophila* has served as a model system to study rDNA CN maintenance in multicellular organisms. The process called rDNA magnification was discovered over 50 years ago in *Drosophila* as a mechanism that recovers rDNA CN of the chromosome that bears particularly low CN of rDNA [15,16]. Although originally found with an unusual chromosome with minimal rDNA CN, rDNA magnification likely represents a mechanism that maintains rDNA CN in wild-type flies, counteracting spontaneous CN reduction during aging, thereby maintaining germline immortality [10,17–19]. A mechanistic model of how rDNA magnification is achieved has emerged through the investigations over decades: it is believed that unequal sister chromatid exchange (USCE) creates two sister chromatids that reciprocally gained and lost

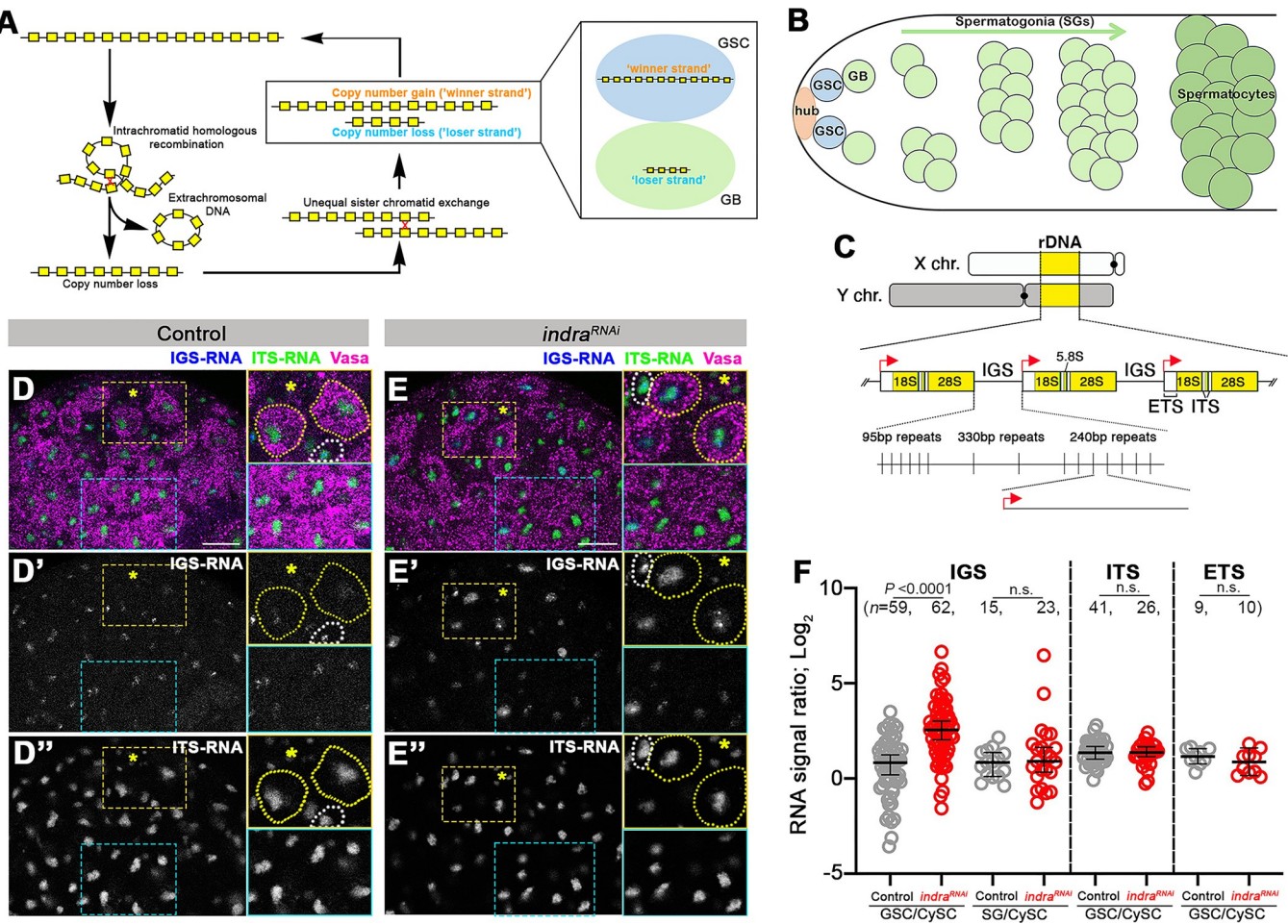

**Fig 1. _indra^RNAi_ leads to upregulation of IGS.** A. Transgenerational dynamics of rDNA copy number (CN). Spontaneous intrachromatid homologous recombination leads to the generation of extrachromosomal rDNA circle and reduction of rDNA CN on the chromosome. To maintain sufficient rDNA CN through generations, the germline must recover rDNA CN, which is achieved by unequal sister chromatid exchange (USCE). During asymmetric divisions of germline stem cells (GSCs), GSCs inherit a sister chromatid that gained rDNA CN after USCE. B. Schematic of Drosophila male germ cell development. GSCs reside at the apical tip of the testis, attaching to the hub cells, the major component of the stem cell niche, to maintain stem cell identity. GSCs divide asymmetrically to produce differentiating daughter cells, gonialblast (GBs), which subsequently proceed to transit-amplifying divisions as spermatogonia before entering meiosis. C. Structure of rDNA. rDNA loci are present on X and Y chromosomes in the genome of _D. melanogaster_. Each rDNA locus contains hundreds of tandemly repeated rDNA units, which contain IGS, ETS, 18S, ITS, 5.8S, and 28S. ETS serves as an RNA polymerase I promoter (red arrow) to cotranscribe downstream rDNA sequences. IGS consists of three subrepeats, 95bp, 330bp, and 240bp. Among the repeats, only 240bp IGS has a promoter sequence duplicated from ETS. D. E. RNA fluorescent _in situ_ hybridization (FISH) to visualize IGS (blue) and ITS (green) transcripts, combined with immunofluorescence staining of Vasa (germ cell; magenta) in control (D) and RNAi-mediated depletion of _indra_ in germ cells (_nos-gal4>UAS-indra^GD9748_, _UAS-Dcr-2_ referred to as _indra^RNAi_, hereafter) (E). Hub cells to mark the stem cell niche are indicated by an asterisk. Yellow insets show enlarged images of the GSC area, while blue insets show enlarged images of the spermatogonia (SGs) area. GSCs are indicated by yellow dotted lines, and cyst stem cells (CySCs) are indicated by white dotted lines in insets. Bar: 10 μm. F. Quantification of IGS, ITS, and ETS expression in GSCs or SGs in control vs. _indra^RNAi_. The signal intensity was normalized by the signal in CySCs, which is genetically wild type (i.e. _indra_ is not depleted). n = number of GSCs and SGs scored. P value, two-tailed Mann-Whitney test. The error bar indicates the median with a 95% confidence interval (CI).

rDNA CN (Fig 1A) [20–22]. USCE appears to specifically occur in male germline stem cells (GSCs), which divide asymmetrically to generate one GSC and one differentiating cell called gonialblast (GB) [18,23] (Fig 1B). During asymmetric GSC division following USCE, the sister chromatid that gained rDNA CN is preferentially inherited by GSCs, through a process called nonrandom sister chromatid segregation (NRSS) [18,19,24]. GSCs' repeated asymmetric divisions enable them to recover rDNA CN through successive rounds of USCE and NRSS [18,19].

Our recent studies identified two critical regulators of rDNA magnification. First, we showed that retrotransposon R2, which inserts into rDNA loci in a DNA sequence-specific manner, is a critical initiator of USCE via its ability to generate DNA double-strand breaks (DSBs) at the rDNA locus [17]. Whereas the function of R2 in maintaining host's rDNA CN represents a striking example of host-transposon mutualism, its unchecked derepression will likely threaten genome integrity. Therefore, its derepression must be carefully controlled by the host, however, the underlying mechanism remains unknown. Second, we discovered a multi-zinc-finger protein Indra that binds rDNA (more specifically, intergenic spacer sequence (IGS)) and is required for rDNA magnification [19]. The mechanism by which Indra regulates rDNA magnification remains unknown.

In this study, we show that Indra is a negative regulator of the IGS transcription. The amount of Indra protein is tightly regulated in GSCs, responding to rDNA CN. Under low rDNA CN condition, Indra amount decreases, leading to upregulation of IGS. We show that it is RNA polymerase II (Pol II) that is responsible for this upregulation of IGS transcription, and this leads to the transcription of R2-inserted rDNA copies. Derepression of R2, in turn, leads to the induction of rDNA DSBs, triggering rDNA magnification. Therefore, Indra is a dose-sensitive regulator of IGS expression, allowing expression of R2 only under the low rDNA CN condition, thereby preventing unnecessary expression of R2. Taken together, we present an integrated model of how rDNA CN is maintained through dynamic transcriptional regulation of R2-inserted vs. -uninserted rDNA copies through differential use of Pol I and II.

## Results

### Indra is a negative regulator of IGS transcription in GSCs

Recently, we identified Indra as an rDNA-binding protein required for rDNA magnification in the *Drosophila* [19]. To explore how Indra regulates rDNA magnification, we examined the impact of *indra* knockdown in the male germline (*nos-gal4>UAS-indra^GD9748*, *UAS-Dcr-2*, referred to as *indra^RNAi* hereafter. The efficiency of *indra^RNAi* was previously shown [19]). Considering that Indra preferentially binds IGS [19], and has multiple Zn-finger domains, we tested whether Indra may regulate IGS expression. Each unit of rDNA contains IGS, ETS (external transcribed spacer), ITS (internal transcribed spacer) and 18S, 5.8S, 28S rRNA (Fig 1C). As observed in budding yeast [25–30], *Arabidopsis* [31], *Xenopus* oocyte [32], *Drosophila* embryo [33], *Drosophila* ovaries [34], and cultured mouse/human cells [35–37], IGS is transcribed in *Drosophila* male germline and somatic cells detected by RNA fluorescence *in situ* hybridization (FISH) (Fig 1D). The expression of ETS and ITS, which are co-transcribed with 18S, 5.8S, and 28S rRNA genes, were also detected by RNA FISH, as expected (Fig 1D).

We found that *indra^RNAi* resulted in striking upregulation of IGS transcription (Fig 1E and 1F), suggesting that Indra is a transcriptional repressor of IGS. Importantly, *indra^RNAi* did not result in any noticeable changes in the level of ETS or ITS (Fig 1E and 1F), revealing Indra's specificity for IGS transcription. This is consistent with Indra's specific binding to IGS, as we reported previously [19]. Additionally, we noted that the impact of *indra^RNAi* on IGS derepression was limited to GSCs: differentiating germ cells (i.e. gonialblasts (GBs), and spermatogonia (SGs)) did not change IGS expression level (Fig 1E and 1F), demonstrating that Indra is a GSC-specific repressor of IGS expression.

### Indra tunes IGS transcription in response to varying rDNA copy number

The above results revealed that Indra negatively regulates IGS expression. In budding yeast, a promoter within IGS, called E-pro, is normally repressed by Sir2, but becomes derepressed when rDNA CN is reduced [27]. In turn, E-pro expression leads to DSB formation at the

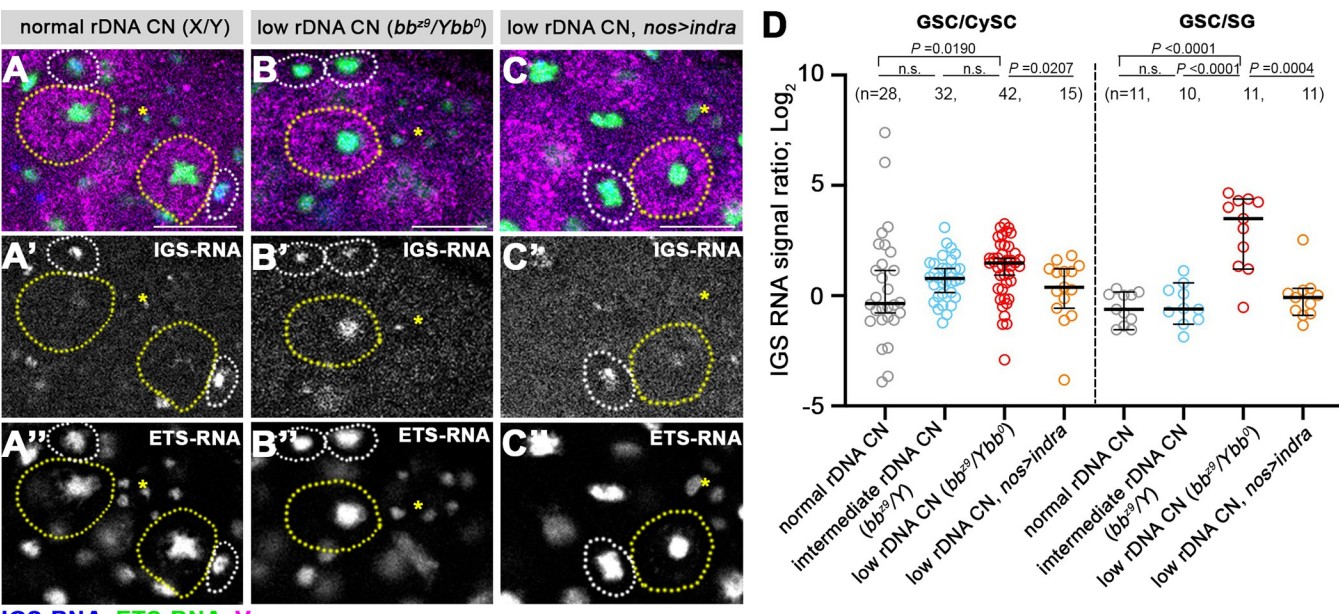

**IGS-RNA ETS-RNA Vasa**

**Fig 2. Low rDNA CN condition induces IGS expression, which is negatively regulated by Indra.** A-C. RNA FISH for IGS (blue) and ETS (green) transcripts combined with immunofluorescence staining for Vasa (magenta) in normal rDNA CN condition (A), low rDNA CN condition (B), or low rDNA CN condition overexpressing Indra (C). Asterisks indicate Hub. GSCs are indicated by yellow dotted circles. CySCs are indicated by white dotted circles. Bar: 10 μm. D. Quantification of IGS transcripts in GSCs compared to CySCs (left) or to SGs (right) under varying rDNA CN. Overexpression of Indra leads to suppression of IGS upregulation observed in low rDNA CN condition. *n* = number of GSCs scored. *P* value, two-tailed Mann-Whitney test. The error bar indicates the median with a 95% CI.

nearby replication fork block site, leading to dislodgement of the cohesin from IGS, triggering rDNA copy number recovery through unequal sister chromatid recombination [27–29].

To test whether IGS expression also responds to rDNA CN in the *Drosophila* male GSCs, we performed RNA FISH for IGS transcripts in animals with varying rDNA CN. In this study, we used three conditions that represent varying rDNA CN by combining various X and Y chromosomes, the chromosomes that harbor rDNA loci in *Drosophila melanogaster*: 1) animals that have critically low rDNA CN on the X chromosome and no rDNA at all on the Y chromosome ($bb^{z9}/Ybb^0$) are known to induce rDNA magnification [15, 16], and this condition is used as the 'low rDNA CN' condition. 2) wild type animals that have normal rDNA CN on both X and Y chromosomes (X/Y) are used as 'normal rDNA CN' condition. 3) animals that have critically low rDNA CN on the X chromosome but have sufficient rDNA CN on the Y chromosome ($bb^{z9}/Y$), which does not induce magnification [17,19], are used as 'intermediate rDNA CN' condition.

Using these conditions, we found that IGS expression was specifically upregulated in GSCs with reduced rDNA CN (Fig 2A–2D). We quantified the relative expression of IGS in GSC compared to CySCs or SGs (GSC/CySC or GSC/SG) under varying rDNA CN conditions. The results show that IGS is upregulated in low rDNA CN condition ($bb^{z9}/Ybb^0$) compared to normal rDNA CN (X/Y) conditions (Fig 2A, 2B and 2D). Intermediate rDNA CN ($bb^{z9}/Y$) exhibited moderate IGS upregulation (Fig 2D), suggesting that IGS expression level negatively correlates with rDNA CN (Fig 2D). Strikingly, we found that overexpression of Indra in germline abrogated IGS upregulation caused by low rDNA CN ($bb^{z9}/Ybb^0$; *nos>indra*), suggesting that Indra is sufficient to repress IGS (Fig 2C and 2D, see S1 Fig for quantification of Indra overexpression, which showed that Indra is ~1.7 times overexpressed in this condition). Importantly, ETS expression did not noticeably change under varying rDNA CN conditions,

or by overexpression of Indra, (Fig 2A"–2C" and 2D). Collectively, these results suggest that only IGS expression changes in response to rDNA CN, and Indra negatively regulates IGS expression.

## Indra protein amount decreases in GSCs in response to low rDNA CN

The result that Indra is a negative regulator of IGS transcription, combined with the result that IGS transcription is upregulated in response to low rDNA CN, we hypothesized that Indra may be downregulated in response to low rDNA CN. To test this possibility, we visualized the Indra protein by immunofluorescence staining of the *Drosophila* testes by using a previously validated anti-Indra antibody [19]. Indra localizes to the nucleolus in interphase, consistent with its affinity to the rDNA sequence (especially IGS) [19]. By comparing relative Indra amount in GSCs vs. SGs (GSC/SG), we found that Indra amount in GSCs decreases under low rDNA CN conditions (Fig 3A–3C). Interestingly, under normal rDNA CN conditions, GSCs have a higher Indra amount compared to SGs, and Indra amount decreases in GSCs under low rDNA CN (Fig 3C). The decrease in Indra amount in GSCs was clear between GSCs and their differentiating daughter, GBs, even when they are still connected by cytoplasmic bridges (Fig 3D–3F). Although GBs/SGs have lower Indra amount than GSCs under normal rDNA CN conditions, it does not seem to lead to higher IGS expression in GBs/SGs. This is probably because Indra-dependent regulation of IGS expression operates only in GSCs as indicated above (Fig 1), and Indra amount or IGS expression is likely not regulated by rDNA CN outside GSCs.

Taken together, these results suggest that the Indra amount in GSCs responds to rDNA CN, and its downregulation is the underlying cause of IGS upregulation under low rDNA CN conditions.

## Indra is a negative regulator of DNA double-strand breaks at rDNA loci

Above results demonstrated that Indra is a negative regulator of IGS expression, and IGS is upregulated under low rDNA CN condition. To understand how these phenomena may be linked to rDNA magnification, we next investigated the relationship between Indra, IGS expression, and rDNA magnification. rDNA magnification is initiated by the DSB formation at the rDNA [17,18,20,38] (Fig 4A). Therefore, we tested whether IGS expression downstream of Indra results in DSB formation. Indeed, we found that *indra*[RNAi] resulted in striking upregulation of DSB formation in GSCs (Fig 4B–4D), suggesting that Indra downregulation is upstream of DSB formation. DSBs observed in *indra*[RNAi] are likely formed at rDNA loci, because mitotic spread of germ cells followed by DNA FISH revealed that X and Y chromosomes are frequently recombined at rDNA loci in *indra*[RNAi], which likely results from excess DNA breaks and exchanges at rDNA loci (S2 Fig).

To test whether DSB formation during rDNA magnification (i.e. low rDNA CN condition) also depends on IGS upregulation, we examined the effect of Indra overexpression under low rDNA CN condition. Whereas DSB increased in GSCs with low rDNA CN as shown previously [18], this increase in DSBs was completely suppressed when Indra was overexpressed ($bb^{z9}/Ybb^0$; *nos>indra*) (Fig 4E), suggesting that Indra is sufficient to suppress DSB formation through its ability to repress IGS expression (Fig 2). As expected from reduced DSBs, Indra overexpression also suppressed rDNA magnification in response to low rDNA CN (Fig 4F, see Methods). Taken together, these results show that IGS upregulation due to reduced Indra amount leads to DSB formation, triggering rDNA magnification, in response to low rDNA CN.

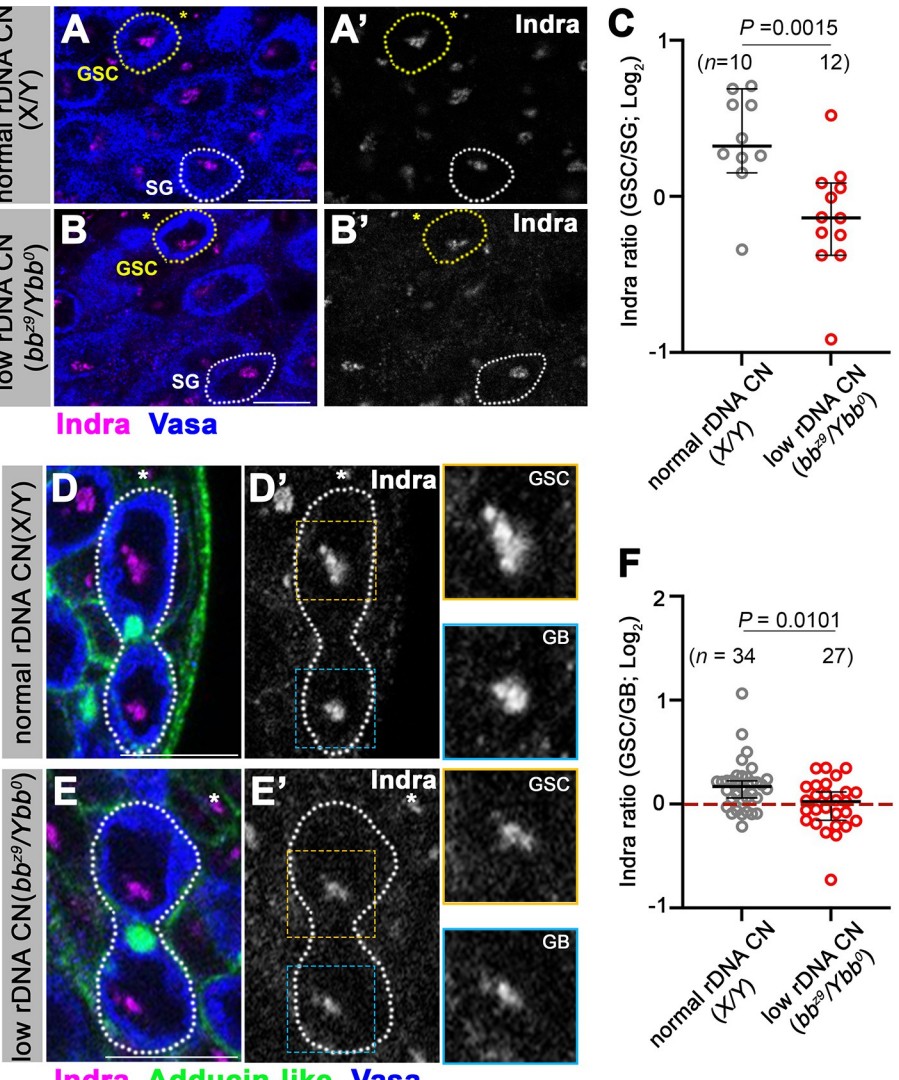

**Fig 3. Indra amount decreases in GSCs under low rDNA CN conditions.** A, B. Immunofluorescence staining of Indra (magenta) and Vasa (blue) under normal rDNA CN (A) or low rDNA CN (B) conditions. The asterisks indicate Hub. GSCs are indicated by yellow dotted lines, SGs by white dotted lines. Bar: 10 μm. C. Quantification of Indra amount in GSCs under normal vs. low rDNA CN conditions (normalized by using Indra signal intensity in SGs). *n* = number of GSCs scored. *P* value, two-tailed Mann-Whitney test. The error bar indicates the median with a 95% CI. D. E. Immunofluorescence staining of Indra (magenta), Adducin-like (the connection between GSC and GB; green), and Vasa (blue) under normal rDNA CN (D) or low rDNA CN (E) conditions in GSCs still connected to GBs. The asterisks indicate Hub. GSC-GB pairs are indicated by white dotted lines. Insets show enlarged images of the GSC vs. GB side of Indra signals in the nucleolus. Bar: 10 μm. F. Quantification of Indra amount in GSCs under normal vs. low rDNA CN conditions (normalized by using Indra signal intensity in connected GBs). *n* = number of GSCs scored. *P* value, two-tailed Mann-Whitney test. The error bar indicates the median with a 95% CI.

## Indra represses R2 expression

How does IGS expression induce DSB formation at rDNA to initiate USCE? In yeast, DSB is created at the replication fork block (RFB) within the IGS when replication forks stall [39]. It is unknown whether *Drosophila* rDNA contains an RFB. However, we recently showed that rDNA-specific retrotransposon R2 can induce DSBs at rDNA in the process of its retrotransposition and that R2 activity is required for inducing rDNA magnification [17]. Therefore, we

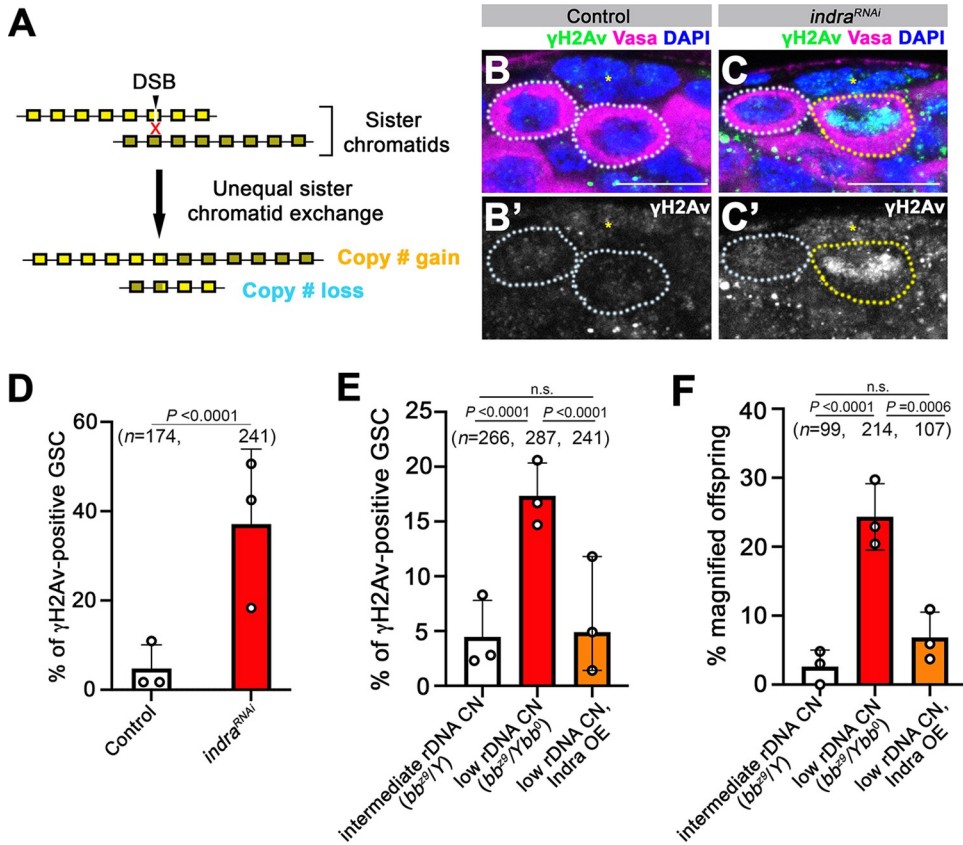

**Fig 4. Indra is a negative regulator of DNA double-strand breaks during rDNA magnification.** A. rDNA magnification is initiated by DSB formation which leads to unequal sister chromatid exchange (USCE). B. C. Apical tip of testes stained for [γH2Av (green), Vasa (magenta), DAPI (blue)] in control (B) and *indra*RNAi (C). The asterisk indicates Hub. γH2Av -positive and -negative GSCs are marked by yellow and white dotted lines, respectively. Bar: 10 μm. D. Frequency of γH2Av-positive GSCs in control vs. *indra*RNAi. *n* = number of GSCs scored. *P* value, two-sided Fisher's exact test. The error bar indicates the mean with standard deviation (SD). E. Frequency of γH2Av-positive GSCs under conditions of normal rDNA CN, low rDNA CN, and low rDNA CN overexpressing Indra. An increase in the frequency of γH2Av-positive GSCs under low rDNA CN conditions is suppressed by Indra overexpression. *n* = number of GSCs scored. *P* values, two-sided Fisher's exact test. The error bar indicates the mean with SD. F. Frequency of rDNA magnification (recovery from *bobbed* phenotype) in the offspring of the fathers of indicated genotypes. rDNA magnification induced by low rDNA CN is suppressed by Indra overexpression. *n* = number of offspring scored. *P* values, two-sided Fisher's exact test. The error bar indicates the mean with SD.

wondered whether IGS expression might be linked to R2 derepression. To test this possibility, we examined R2 expression by RNA FISH and found that the frequency of R2-positive GSCs is dramatically increased in *indra*RNAi GSCs (Fig 5A–5C), suggesting that Indra represses R2 expression. Moreover, upregulation of R2 in response to low rDNA CN (Fig 5D) [17] was suppressed by overexpression of Indra (*bb*z9/*Ybb*0, *nos>indra*) (Fig 5D), suggesting that Indra is sufficient to repress R2.

The results thus far suggest that IGS expression is linked to R2 expression, both of which are under negative regulation by Indra. Because R2 does not have own promoter, its expression relies on read-through transcription of rDNA copies in which R2 is inserted [40]. However, R2-inserted rDNA copies are normally repressed [17,41]. Thus, a possible model to explain the link between IGS expression and R2 expression is the following: normally, R2-inserted rDNA is repressed, involving the function of Indra. However, under low rDNA CN conditions, the Indra amount decreases, leading to the derepression of rDNA copies

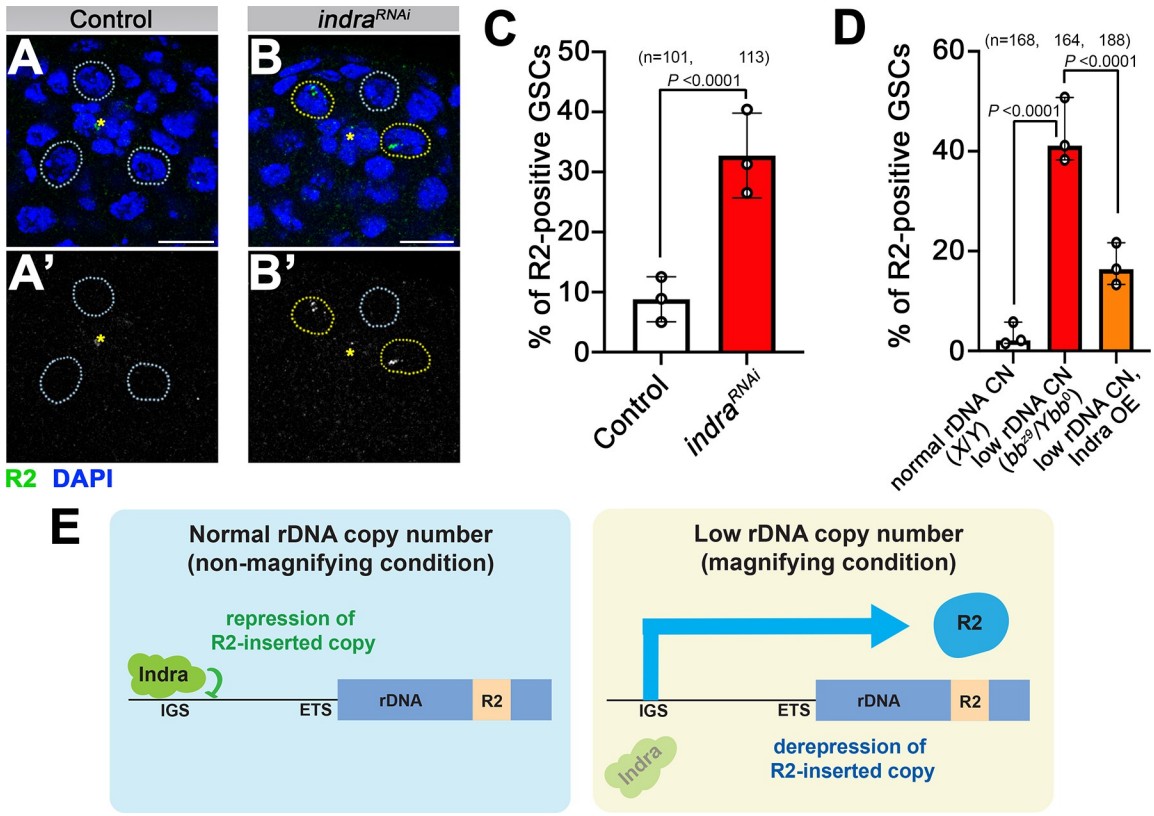

**Fig 5. Indra is a negative regulator of R2 expression during rDNA magnification.** A. B. RNA FISH for R2 (green), counterstained with DAPI (blue), in control (A) and *indra^RNAi* (B) GSCs. GSCs with R2 expression are indicated by yellow dotted circles, and GSCs without R2 expression are indicated by cyan dotted circles. The asterisk indicates Hub. Bar: 10 μm. C. Frequency of R2 positive GSCs in control vs. *indra^RNAi*. n = number of GSCs scored. P value, two-sided Fisher's exact test. The error bar indicates the mean with SD. D. Frequency of R2 positive GSCs under normal rDNA CN condition, low rDNA CN condition, low rDNA CN condition overexpressing Indra. Indra expression suppresses R2 upregulation under low rDNA CN. n = number of GSCs scored. P value, two-sided Fisher's exact test. The error bar indicates the mean with SD. E. Model of R2 derepression by IGS expression under low rDNA CN. Although R2-inserted rDNA copies are normally repressed, reduction in Indra protein amount under low rDNA CN condition leads to IGS expression, leading to transcription of R2. This may be a triggering event to induce rDNA magnification.

inserted with R2, leading to R2 expression (Fig 5E). This in turn may lead to rDNA magnification.

## RNA polymerase II is recruited to the periphery of the nucleolus in response to low rDNA CN

How does Indra regulate IGS expression? It has been shown that IGS can be transcribed by RNA polymerase II (Pol II) [36,42–44]. Strikingly, we found that Pol II is recruited to the nucleolar periphery under low rDNA CN conditions in GSCs (Fig 6A–6D). Pol II is mostly excluded from the nucleolus, the site of rDNA transcription, in animals with normal rDNA CN (Fig 6A and 6B). However, prominent Pol II localization was observed at the periphery of the nucleolus in animals with low rDNA CN (Fig 6C and 6D). Importantly, localization change of Pol II to the nucleolus was most noticeable in GSCs, and less striking in differentiating germ cells (Fig 6E–6I). These results indicate that Pol II is recruited to the nucleolus to express IGS in GSCs under low rDNA CN conditions. Moreover, *indra^RNAi* resulted in Pol II accumulation at the nucleolus specifically in GSCs but not in SGs (Fig 6J–6R), mirroring the observation

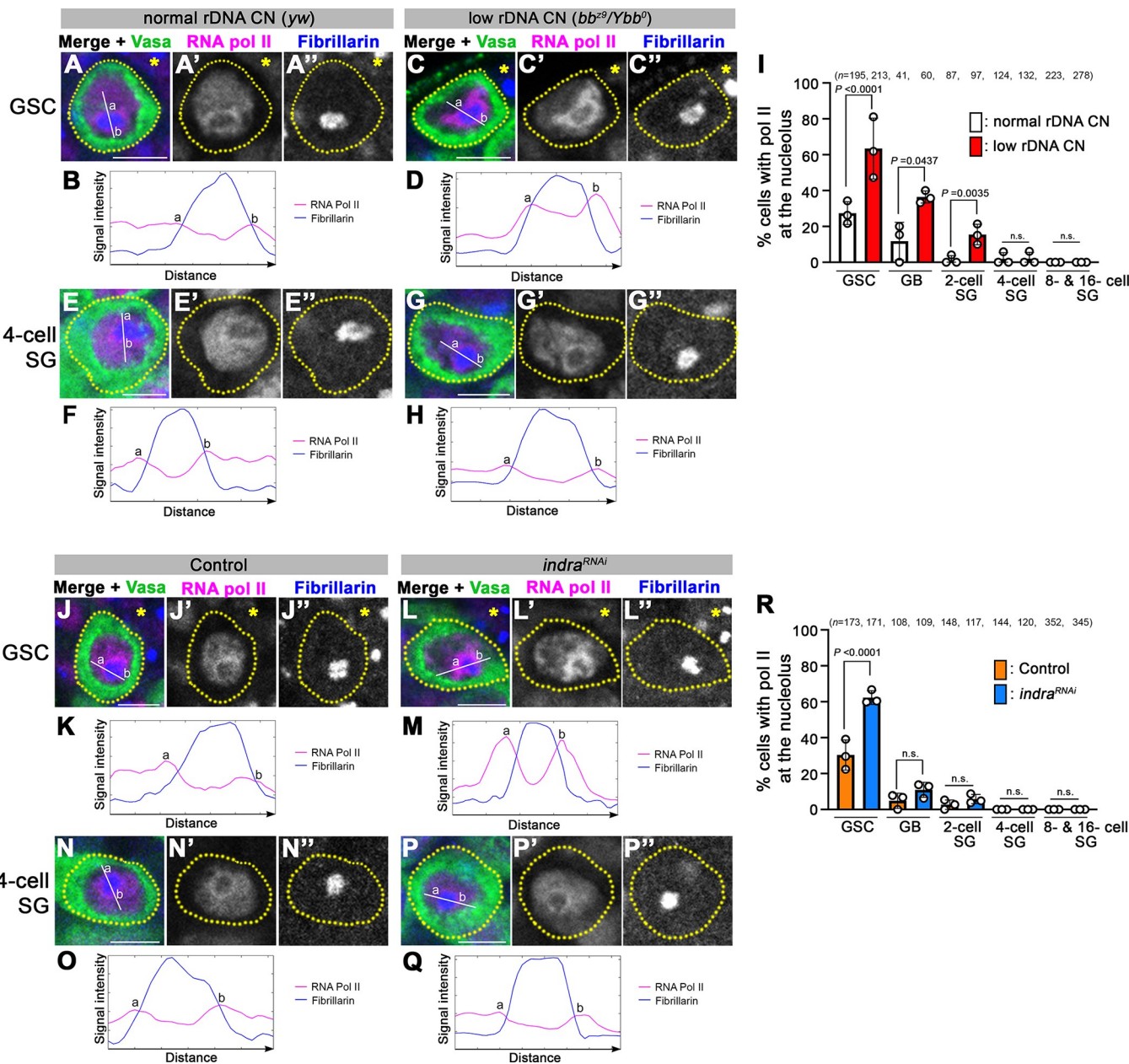

**Fig 6. RNA polymerase II localizes to the periphery of nucleolus under low rDNA CN condition, which is negatively regulated by Indra.** A-D. RNA polymerase II localization in GSCs under normal rDNA CN (A) and low rDNA CN (C) conditions. Immunofluorescence staining of RNA polymerase II (CTD4H8, magenta), Fibrillarin (blue, nucleolus marker), and Vasa (green, germ cells). Signal intensity of Pol II (magenta line) and Fibrillarin (blue line) across nucleolus under normal rDNA CN (B) and low rDNA CN (D) conditions is shown. The asterisk indicates Hub. Bar: 5 μm. E-H. RNA polymerase II localization in SGs is unchanged under normal rDNA CN (E) and low rDNA CN (G) conditions. Signal intensity of Pol II (magenta line) and Fibrillarin (blue line) across nucleolus under normal rDNA CN (F) and low rDNA CN (H) conditions is shown. I. Frequencies of cells with RNA polymerase II at the nucleolar periphery at different stages of germ cell development in indicated genotypes. $n$ = number of cells scored. $P$ value, two-sided Fisher's exact test. The error bar indicates the mean with SD. J-M. RNA polymerase II localization in GSCs from control (J) and $indra^{RNAi}$ (L) animals (with normal rDNA CN). Signal intensity of Pol II (magenta line) and Fibrillarin (blue line) across nucleolus in control (K) and $indra^{RNAi}$ (M) is shown. N-Q. RNA polymerase II localization in SGs from control (N) and $indra^{RNAi}$ (P) animals (with normal rDNA CN). Signal intensity of Pol II (magenta line) and Fibrillarin (blue line) across nucleolus in control (O) and $indra^{RNAi}$ (Q) is shown. R. Frequencies of cells with RNA polymerase II at the nucleolar periphery at different stages of germ cell development in control vs. $indra^{RNAi}$. $n$ = number of cells scored. $P$ value, two-sided Fisher's exact test. The error bar indicates the mean with SD.

under low rDNA CN conditions. These results suggest that Indra represses the recruitment of Pol II to the IGS/nucleolus, thereby inhibiting Pol II-mediated transcription of IGS.

### IGS and R2 are transcribed by RNA polymerase II in response to low rDNA CN

Based on the results described above, we hypothesized that Pol II-mediated IGS transcription might trigger expression of R2, either directly transcribing through rDNA inserted with R2 or by activating Pol I-dependent promoter (ETS) upstream of R2-inserted rDNA copies. To test whether Pol II is responsible for R2 expression (directly or indirectly), we used Pol II inhibitor, α-amanitin, in *ex vivo* testis culture. Testes from animals with normal rDNA CN exhibited minimal expression of IGS and R2 (Fig 7A), and those from animals with low rDNA CN exhibited upregulation of IGS and R2 (Fig 7B). This upregulation of IGS and R2 in the testes with low rDNA CN was completely abolished upon addition of α-amanitin (Figs 7C–7E and S3), demonstrating that transcription of IGS and R2 are indeed dependent on Pol II. Importantly, α-amanitin treatment did not noticeably impact 5S rRNA or ETS expression (Fig 7B''' and 7C''', 7F and S3), consistent with its being transcribed by Pol III or Pol I, respectively. Moreover, similar to low rDNA condition, IGS and R2 upregulation in *indra^{RNAi}* was completely repressed by α-amanitin treatment, without affecting the expression level of ETS (Fig 7G–7K), demonstrating that Indra is a negative regulator of Pol II-dependent IGS/R2 transcription.

Taken together, we conclude that Pol II is recruited to the nucleolar periphery under low rDNA CN conditions to transcribe IGS, which in turn activate the transcription of R2, leading to rDNA magnification.

### Discussion

Maintenance of genome integrity is of the highest importance in the germline, the immortal lineage that transmits the genome in an eternal cycle of life. rDNA is one of the most vulnerable loci in the genome, thus its maintenance is of parament importance. In this study, we characterized the function *indra*, a gene we previously discovered to be required for rDNA magnification, the process of recovering rDNA CN to counteract spontaneous CN loss [19]. Our results collectively provide a mechanistic model of how *indra* is involved in the regulation of rDNA magnification. The present study shows that Indra functions as a negative regulator of IGS expression. Under low rDNA CN condition, Indra protein amount decreases, which leads to derepression of IGS. This in turn leads to derepression of R2, the retrotransposon required for rDNA magnification, leading to DSB formation that triggers USCE-mediated rDNA CN recovery. This study provides an integrated model of rDNA CN maintenance. Based on our findings described in this study, we propose the following model (Fig 7L): expression of rDNA copies inserted with R2 is normally repressed by Indra which binds to IGS upstream of ETS. When rDNA CN is reduced, Indra amount decreases and Pol II is recruited to the Indra-free IGS promoter to transcribe the R2-inserted rDNA copies, leading to R2 derepression specifically under low rDNA CN conditions. R2 derepression then leads to USCE, resulting in rDNA magnification. In contrast, R2-uninserted rDNA copies are normally transcribed by Pol I to produce ribosomes, and this is not influenced by rDNA CN. Critically, our model proposed here implies that the regulation of rDNA magnification converges on the regulation of Indra protein amount. How the cells sense rDNA copy number to regulate Indra protein amount awaits future investigation.

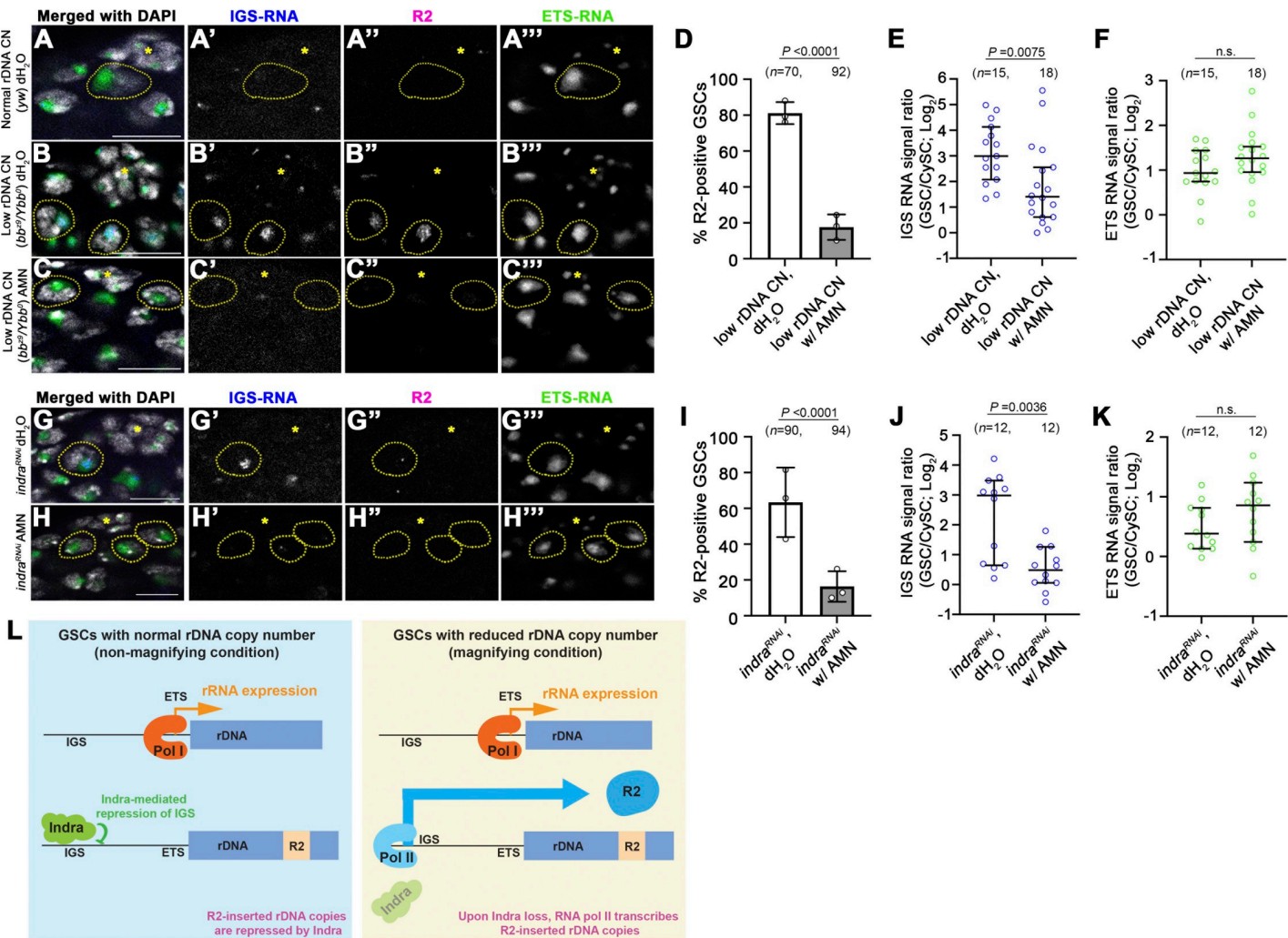

**Fig 7. RNA polymerase II transcribes IGS and R2 under low rDNA CN conditions.** A-C. *ex vivo* culture of *Drosophila* testis with normal rDNA CN (A), low rDNA CN (B), and low rDNA CN treated with α-amanitin (AMN). *in situ* hybridization for IGS (blue) and R2 (magenta) and ETS (green). Bar: 10 μm. D. Frequency of R2-positive GSCs in low rDNA CN testes with or without α-amanitin (Pol II inhibitor). *n* = number of cells scored. *P* value, two-sided Fisher's exact test. The error bar indicates the mean with SD. E. Quantification of IGS transcripts in GSCs compared to CySCs in low rDNA CN testes with or without α-amanitin. *n* = number of GSCs scored. *P* value, two-tailed Mann-Whitney test. The error bar indicates the median with a 95% CI. F. Quantification of ETS transcripts in GSCs compared to CySCs in low rDNA CN testes with or without α-amanitin. *n* = number of GSCs scored. *P* value, two-tailed Mann-Whitney test. The error bar indicates the median with a 95% CI. G, H. *ex vivo* culture of *Drosophila* testis in *indra^RNAi^*, treated with dH₂O or α-amanitin. *in situ* hybridization for IGS (blue) and R2 (magenta) and ETS (green). Bar: 10 μm. I. Frequency of R2-positive GSCs in *indra^RNAi^* with or without α-amanitin. *n* = number of cells scored. *P* value, two-sided Fisher's exact test. The error bar indicates the mean with SD. J. Quantification of IGS transcripts in GSCs compared to CySCs in *indra^RNAi^* testes with or without α-amanitin. *n* = number of GSCs scored. *P* value, two-tailed Mann-Whitney test. The error bar indicates the median with a 95% CI. K. Quantification of ETS transcripts in GSCs compared to CySCs in *indra^RNAi^* testes with or without α-amanitin. *n* = number of GSCs scored. *P* value, two-tailed Mann-Whitney test. The error bar indicates the median with a 95% CI. L. Model of R2 derepression by IGS expression under low rDNA CN. Expression of R2-inserted rDNA copies is mediated by Pol II from the IGS promoter, which is normally repressed by Indra under normal rDNA CN conditions. Reduction of Indra protein amount under low rDNA CN condition leads to derepression of IGS and R2. In contrast, R2-uninserted rDNA copies are not subject to this regulation and are transcribed by Pol I.

## Transcriptional regulation of R2-inserted rDNA copies by Indra and RNA polymerase II

Because R2 is a retrotransposon, whose unchecked expression threatens genome integrity, R2 is normally silenced [17,41]. As R2 lacks its promoter and is expressed via read-through of rDNA transcription in which R2 is inserted, R2 repression is achieved by repression of R2-inserted rDNA copies [41]. It remained unknown how cells can distinguish R2-inserted vs.

-uninserted rDNA copies such that cells can selectively express uninserted copies for ribosome biogenesis, whereas R2-inserted copies can be activated only when necessary (i.e. low rDNA CN condition). Our results hint at how R2-inserted copies can be specifically activated during rDNA magnification: R2-inserted copies are normally repressed by Indra at IGS promoter (Fig 7E). Upon reduction in Indra amount under low rDNA CN conditions, IGS becomes upregulated, leading to expression of R2-inserted copies. Pol II may transcribe the entire copy of rDNA inserted with R2, alternatively, Pol II-dependent IGS expression may activate Pol I-dependent transcription of rDNA copies inserted with R2.

Localization of Pol II to the nucleolar periphery and its importance in rDNA/ribosome biology has been reported previously in human cells [36,43,44]. However, it remained unknown why and how Pol II is required, and what role Pol II plays in rDNA biology distinct from Pol I. Our study demonstrating that Pol II may be utilized to transcribe R2-inserted copies provides an interesting example of how two promoters for two distinct RNA polymerases (IGS for Pol II, ETS for Pol I) can be utilized to differentially regulate rDNA copies (e.g. R2-inserted vs. -uninserted).

However, it should be noted that, while many species have rDNA-specific retrotransposons, others do not [41,45]. For example, budding yeast lacks such transposons. Then, is the mechanism described in this study irrelevant for rDNA CN recovery in the species that lack rDNA-specific retrotransposons? We speculate that the differential use of Pol I vs. II may play a similarly critical role even in species that do not rely on retrotransposons. For example, in budding yeast, Pol II-mediated IGS (called E-pro) transcription plays a critical role in rDNA CN recovery [27]. Thus, even if cells do not have to differentiate distinct copies of rDNAs (R2-inserted vs. -uninserted), the use of Pol II for the expression of IGS may allow the expression of IGS to be reserved only for 'special occasion' (i.e. low rDNA CN condition), without the danger of Pol I accidentally transcribing IGS due to its recruitment to proximity (ETS promoter).

### Unique characteristics of GSCs and the mechanism that monitors rDNA CN

The data presented in this study revealed unique characteristics of GSCs. Many responses that occur under low rDNA CN conditions were GSC specific: It was predominantly GSCs that upregulated IGS, downregulated Indra protein, and exhibited Pol II recruitment to the perinucleolar region under low rDNA CN conditions. Moreover, the impact of *indra*[RNAi] was also mostly limited to GSCs: both dramatic upregulation of IGS transcription and Pol II recruitment to the nucleolar periphery in *indra*[RNAi] were specific to GSCs. These results reveal unique characteristics of GSCs, consistent with our recent report that rDNA magnification operates specifically in GSCs [18]. We speculate that additional factors that regulate IGS expression may be limited to GSCs, making other cells (GBs and SGs) insensitive to rDNA CN changes and *indra* manipulation.

Whereas the present study demonstrates that Indra protein amount responds to rDNA CN, leading to regulation of rDNA magnification, it remains unknown what regulates Indra protein amount. The present study places the regulation of Indra protein amount upstream of other steps during rDNA magnification (e.g. IGS expression, R2 expression, and DSB formation). However, the most critical, unresolved question is how cells (most likely GSCs) monitor rDNA CN, translating this information into Indra protein amount. It awaits future investigations to understand this most critical question of how Indra protein amount is regulated by rDNA CN.

### Indra and nonrandom sister chromatid segregation

Indra was originally discovered as a gene, whose depletion compromises nonrandom sister chromatid segregation (NRSS) [19]. NRSS is a process where two sister chromatids, which are

supposed to be identical copies of each other, are somehow differentiated and segregated non-randomly during cell division. NRSS has been often studied in the context of the 'immortal strand hypothesis' which proposed that long-living cells must retain old DNA strands to avoid accumulation of replication-induced mutations [46]. However, we found no evidence for 'whole genome level' NRSS in *Drosophila* male GSCs, making it unlikely that NRSS operates to retain the 'immortal strands' [47]. Instead, we unexpectedly found that *Drosophila* male GSCs exhibit chromosome-specific NRSS: instead of the whole genome, only the X and Y chromosomes exhibit NRSS [24]. Subsequently, we showed that NRSS of the X and Y chromosomes depends on rDNA loci, and provided evidence that NRSS allows GSCs to select the sister chromatid that gained rDNA CN upon USCE [19].

Based on the data that *indra*[RNAi] compromises NRSS [19], we initially hypothesized that Indra is involved in the choice of sister chromatids during GSC divisions. However, the present study shows that Indra is a negative regulator of USCE, rather than being involved in the choice of sister chromatids. We found that *indra*[RNAi] leads to excess sister chromatid exchanges (S4 Fig), consistent with Indra's role in repressing USCE as discovered in this study. Thus, these results suggest that Indra is not a direct regulator of NRSS: instead, *indra*[RNAi] causes too many USCEs, thereby randomizing the choice of sister chromatids to be inherited by GSCs. Accordingly, whereas the present study provides a mechanistic model of how Indra is involved in the regulation of rDNA magnification, it leaves the mechanism of sister chromatid choice elusive.

## Materials and methods

### Fly husbandry and strains

All fly stocks were raised on the standard Bloomington medium at 25°C containing 0.15% Tegosept as an antifungal (no propionic acid was added). The following fly stocks were used: *UAS-Dcr-2* (BDSC24650), *UAS-indra*[TRiP.HMJ30228] (BDSC63661), *bb*[158], *y*[1]*/Dp(1;Y)y*[+]*/C* [1]*; *ca*[1] *awd*[K] (BDSC3143), and *FM6/C* [1]*DX, y\* f*[1]*/Y* (BDSC784) were obtained from the Bloomington Drosophila Stock Center. *UAS-indra*[GD9748] (v20839) was obtained from the Vienna Drosophila Resource Center. *y*[1] *eq*[1]*/Df(YS)bb*[−] (DGRC101260) was obtained from the Kyoto Stock Center. *UAS-indra-3HA* (F000633) was obtained from the Zurich ORFeome Project (FlyORF). *nos-gal4* [48], *UAS-Upd* [49], *tub-gal80*[ts] [50], and *nos-gal4* without VP16 [51] have been previously described.

Since *nos-gal4>UAS-indra*[TRiP.HMJ30228] causes severe germ cell loss due to a high RNAi efficiency [19], we used a weaker RNAi line (*nos-gal4>UAS-indra*[GD9748] and *UAS-Dcr-2*) and conditional knockdown system by temperature-sensitive GAL4 inhibition (*tub-gal80*[ts]; *nos-gal4ΔVP16>UAS-indra*[TRiP.HMJ30228]) in this study. We have shown these conditions deplete Indra protein but leave sufficient numbers of GSCs for analysis [19].

### Immunofluorescence staining

Immunofluorescence staining was conducted as described previously [19]. Briefly, *Drosophila* adult testes were dissected in phosphate-buffered saline (PBS), transferred to 4% formaldehyde in PBS, and fixed for 30 min. The testes were then washed in PBST (PBS containing 0.1% TritonX-100) for at least 30 min, followed by incubation with primary antibody in 3% bovine serum albumin (BSA) in PBST at 4°C overnight. Samples were washed for 60 min (3x washes for 20 min each) in PBST, incubated with secondary antibody in 3% BSA in PBST at 4°C overnight, washed as above, and mounted in VECTASHIELD with 4′,6-diamidino-2-phenylindole (DAPI; Vector Laboratories, Burlingame, CA).

Primary antibodies used: mouse anti-adducin-like [1:20; 1B1, developed by H. D. Lipshitz, obtained from Developmental Studies Hybridoma Bank (DSHB)], rat anti-vasa (1:20; developed by A. C. Spradling and D. Williams, obtained from DSHB), guinea pig anti-Indra (1:500) [19], rabbit anti-vasa (1:200; d-26, Santa Cruz Biotechnology) rabbit anti-γ-H2AvD pS137 (1:200; Rockland), mouse anti-RNA polymerase II (1:500; CTD4H8; Upstate), and rabbit anti-Fibrillarin (1:200; Abcam). Alexa Fluor–conjugated secondary antibodies (Life Technologies) were used at a dilution of 1:200. Images were taken on a Leica TCS SP8 confocal microscope with a 63x oil immersion objective [numerical aperture (NA) = 1.4] and processed using Adobe Photoshop and ImageJ software.

## RNA fluorescent *in situ* hybridization

RNA FISH was conducted as described previously [52]. Briefly, testes from 2–3 day-old flies were dissected in 1x PBS and fixed in 4% formaldehyde in 1x PBS for 30 minutes. Then testes were washed briefly in PBS and permeabilized in 70% ethanol overnight at 4˚C. Testes were briefly rinsed with the wash buffer (2x saline-sodium citrate (SSC), 10% formamide) and then hybridized overnight at 37˚C in the hybridization buffer (2x SSC, 10% dextran sulfate (sigma, D8906), 1 mg/mL *E. coli* tRNA (sigma, R8759), 2 mM Vanadyl Ribonucleoside complex (NEB S142), 0.5% BSA (Ambion, AM2618), 10% formamide). Following hybridization, samples were washed three times in the wash buffer for 20 minutes each at 37˚C and mounted in VEC-TASHIELD with DAPI (Vector Labs). All solutions used for RNA FISH were RNase-free. Images were taken on a Leica TCS SP8 confocal microscope with a 63x oil immersion objective [numerical aperture (NA) = 1.4] and processed using Adobe Photoshop and ImageJ software. Probe sequences are listed in the S1 Table. R2 Stellaris FISH probe set was designed and synthesized by LGC Biosearch Technologies and used previously [17].

## RNA fluorescent *in situ* hybridization with immunofluorescence staining

RNA FISH combined with immunofluorescence staining was conducted as described previously [53]. Briefly, to combine immunofluorescence staining with RNA FISH, testes from 2–3 day-old flies were dissected in 1x PBS and fixed in 4% formaldehyde in 1x PBS for 30 min. Then testes were washed briefly in PBS and permeabilized in 70% ethanol overnight at 4˚C. Testes were then washed with 1x PBS and blocked for 30 min at 37˚C in the blocking buffer (1x PBS, 0.05% BSA, 50 μg/ml *E. coli* tRNA, 10 mM Vanadyl Ribonucleoside complex, and 0.2% Tween-20). Primary antibodies were diluted in the blocking buffer and incubated at 4˚C overnight. The testes were washed with 1x PBS containing 0.2% Tween-20, incubated in the blocking buffer for 5 min at 37˚C, and then in the blocking buffer containing secondary antibodies at 4˚C overnight. Then, testes were washed with 1x PBS containing 0.2% Tween-20 and fixed with 4% formaldehyde in 1x PBS for 10 min before proceeding with the RNA FISH method, starting from the brief rinse with the wash buffer. Probe sequences are listed in the S1 Table.

## DNA fluorescent *in situ* hybridization on mitotic chromosome spread

DNA FISH on mitotic chromosome spread was conducted as described previously [54]. *Drosophila* testes were squashed, similar to brain squash as previously described [55]. Briefly, testes were dissected in 1x PBS, transferred into a drop of 0.5% sodium citrate on the superfrost plus slide glass (Thermo Fisher Scientific) for 5–10 min, then fixed in 45% acetic acid/2.2% formaldehyde for 4–5 min. Fixed tissues were firmly squashed by placing a cover slip onto the slide glass and applying pressure onto it. The slides were then submerged in liquid nitrogen. After removing the coverslip, the samples on the slides were dehydrated in 100% ethanol for at least

5 min at room temperature, and let dry. Hybridization mix (50% formamide, 2x SSC, 10% dextran sulfate) with 1 μM each probe was applied directly to the slide. The sample was then covered by a coverslip and DNA was heat denatured at 95°C for 2 min. The slides were then incubated in a humid chamber for 16 hours at room temperature. Then the slides were washed 3 times for 15 min in 0.2x SSC, and mounted with VECTASHIELD with DAPI (Vector Labs). Images were taken on a Leica TCS SP8 confocal microscope with a 63x oil immersion objective [numerical aperture (NA) = 1.4] and processed using Adobe Photoshop and ImageJ software. Probe sequences are listed in the S1 Table.

### Chromosome orientation fluorescence *in situ* hybridization (CO-FISH) on mitotic chromosome spread

CO-FISH on mitotic chromosome spread was conducted by modifying the CO-FISH protocol as described previously [24]. 2–3 day-old *nos-gal4>UAS-Upd* (control) and *nos-gal4>UAS-Upd, UAS-indra$^{GD9748}$*, and *UAS-Dcr-2* (*indra$^{RNAi}$*) flies were fed with BrdU-containing apple juice agar for 16–18 hours. Because the average GSC cell cycle length is ~12 hours, most GSCs undergo a single S phase in the presence of BrdU under this condition. *Upd*-expressing testes were used to enrich GSCs. Mitotic spread was prepared as described above, except for using the fixative with lower concentration of acetic acid (13% acetic acid/4% formaldehyde) for 4–5 min, as we found that the use of high acetic acid concentration breaks DNA during sample preparation, leading to hybridization signal without BrdU incorporation. The slides were rehydrated in PBS for 5 min. Then, the slides were incubated with RNase A for 15 min at 37°C and briefly rinsed with PBST. Subsequently, the slides were fixed in 4% formaldehyde in PBS, followed by one PBS rinse. The slides were dehydrated in 75, 85, and then 100% ice-cold ethanol for 2 min each. After the slides were completely air-dried, they were stained with 0.5 μg/ml Hoechst 33258 in 2x SSC for 15 min at room temperature and briefly washed twice in 2x SSC. 200 μl of 2x SSC was added to the slide, and it was covered by a cover slip, and then exposed to ultraviolet light in the CL-1000 Ultraviolet Crosslinker (UVP; wavelength: 365 nm; calculated dose: 5400 J/m$^2$). The slides were briefly rinsed in 2x SSC, then in distilled water and air-dried. Then, the slides were treated with 3 U/μl exonuclease III (New England Biolabs) in 1x NEB cutsmart buffer and incubated at 37°C for 15 min to digest nicked BrdU-positive strands, followed by a wash with 2x SSC twice. The slides were treated in 50% formamide/2x SSC for 10 min at room temperature and immediately dehydrated in ice-cold ethanol series (75, 85, 100% ethanol, 2 min each). Hybridization mix (50% formamide, 2x SSC, 10% dextran sulfate) with 1.5–3 μM IGS probes was denatured at 72°C for 5 min and immediately cooled down on ice for 5 min before being applied to the sample for hybridization. After the hybridization mix was applied directly to the slides, the sample was covered by a coverslip and incubated in a humid chamber for 16 hours at 37°C. Then, the slides were washed once in 50% formamide/2x SSC, 3 times in 2x SSC, and mounted with VECTASHIELD with DAPI (Vector Labs). Images were taken using Leica TCS SP8 confocal microscope with a 63x oil immersion objective (NA = 1.4) and processed using Adobe Photoshop software. Probe sequences are listed in S1 Table. All reagents contained 1 mM EDTA except for one step before enzymatic reaction.

### Image quantification

Fluorescence quantification was carried out with merged Z stacks using ImageJ "Sum of pixel intensity (RawIntDen)". Images were taken using Leica SP8 confocal microscope, using the setting to detect saturation to ensure that acquired signals were not saturated. Also, to avoid

the effect of signal intensity changes (i.e. reduction in signal intensity in deeper focal places), we scored cell pairs only when two cells were found within the same Z plane.

## Magnification assay

Magnification assay was conducted as described previously [19]. The magnification is known to be induced in the $bb^{z9}/Ybb^0$ males ($bb^{z9}$ X chromosome with an insufficient rDNA copy number and $Ybb^0$ Y chromosome that carries no rDNA) [17,19]. These males are crossed to $bb^{158}$/FM6 females and the resultant $bb^{z9}/bb^{158}$ daughters were selected for assay. If magnification had occurred, $bb^{z9}/bb^{158}$ daughters exhibited wild-type cuticles, whereas they showed *bobbed* cuticle if magnification had not occurred. The frequency of magnification was calculated as % daughters with wild-type cuticle among total female progeny (of the genotype $bb^{z9}/bb^{158}$).

## *Ex vivo* treatment of *Drosophila* testis

Testes from 2–3 day-old flies were dissected and transferred to Schneider's insect medium (Gibco) with or without 50 ng/µl α-amanitin. After 2 hours of incubation at room temperature, testes were fixed in 4% formaldehyde in 1x PBS for 30 minutes and processed for RNA fluorescent *in situ* hybridization.

## Statistical analysis

For comparison of RNA signal ratio between control and $indra^{RNAi}$ flies in Fig 1F, IGS RNA signal ratio among flies with varying rDNA CN in Fig 2D, Indra signal ratio between normal rDNA CN flies and low rDNA CN flies in Fig 3C and 3F, RNA signal ratio with or without α-amanitin in Fig 7E, 7F, 7J, and 7K, and Indra signal ratio with or without Indra overexpression in S1 Fig two-tailed Mann-Whitney tests determined significance. Other than these, significance was determined by two-sided Fisher's exact tests.

## Supporting information

**S1 Fig. Quantification of Indra overexpression.** Indra amount in GSCs relative to CySCs under low rDNA condition without or with Indra overexpression in GSCs. Because CySCs do not express Indra transgene, it serves as an appropriate denominator to determine the level of Indra overexpression in GSCs. $n$ = number of GSCs scored. $P$ value, two-tailed Mann-Whitney test. The error bar indicates the median with a 95% CI.
(TIF)

**S2 Fig. X and Y chromosomes undergo frequent recombination in $indra^{RNAi}$ early germ cells.** A, B) DNA FISH on the mitotic chromosome spread from control (A) or $indra^{RNAi}$ (B) early germ cells. FISH probes: Alexa488-IGS (rDNA loci on X and Y chromosomes, green); Cy3-$(TAGA)_8$, Cy3-359 (X chromosome, red); Cy5-$(AATAC)_6$, Cy5-$(AATAAAC)_6$ (Y chromosome, blue). Bar: 2.5µm. C) Frequency of early germ cells that exhibit recombination at rDNA upon *indra* knockdown. ($nos$-$gal4$>$UAS$-$indra^{GD9748}$, $UAS$-$Dcr$-$2$, and $tub$-$gal80^{ts}$, $nos$-$gal4\Delta VP16$>$UAS$-$indra^{TRiP.HMJ30228}$). $n$ = number of mitotic spreads of early germ cells scored. $P$ values, two-sided Fisher's exact test. The error bar indicates the mean with SD.
(TIF)

**S3 Fig. IGS upregulation under low rDNA CN condition is eliminated by Pol II inhibitor α-amanitin.** A, B) in situ hybridization of IGS (A', B'), 5S rRNA (A", B"), and ETS (A'", B'") transcripts under low rDNA conditions treated with dH2O (A-A"') or α-amanitin (AMN,

B-B'''). IGS was specifically sensitive to Pol II inhibition by α-amanitin, whereas 5S rDNA (transcribed by Pol III) and ETS (transcribed by Pol I) were not affected. Bar: 10 μm.
(TIF)

**S4 Fig. rDNA loci exhibit excess sister chromatid exchanges in *indra*^RNAi germline stem cells.** A-C) CO-FISH on mitotic spread of germline stem cells (GSCs) in control (A, B) and *indra*^RNAi (C). In control, 'blue strand' and 'red strand', representing each sister chromatid, were juxtaposed and no sister chromatid exchange was observed in 73% of GSCs (A). In the remaining 27% of cases (B), one of rDNA loci (X or Y) exhibited one sister chromatid exchange. In *indra*^RNAi, we often (67%) observed multiple sister chromatid exchanges. This may also involve homologous recombination between X and Y rDNA loci, as indicated by the data shown in S1 Fig. Bar: 5μm. Note that *upd*-overexpression condition was used to enrich GSCs.
(TIF)

**S1 Table. Probe sequences for RNA FISH, DNA FISH, and CO-FISH.**
(DOCX)

**S1 Raw Data. Containing raw data used to generate graphs presented in the manuscript.**
(XLSX)

## Acknowledgments

We thank the Bloomington Stock Center, the Vienna Drosophila Resource Center, and Developmental Studies Hybridoma Bank for reagents, the members of the Yamashita lab, and Dr. Scott Hawley for discussions and comments on the manuscript.

## Author Contributions

**Conceptualization:** George J. Watase, Yukiko M. Yamashita.

**Formal analysis:** George J. Watase.

**Funding acquisition:** George J. Watase, Yukiko M. Yamashita.

**Investigation:** George J. Watase.

**Supervision:** Yukiko M. Yamashita.

**Validation:** George J. Watase, Yukiko M. Yamashita.

**Visualization:** George J. Watase, Yukiko M. Yamashita.

**Writing – original draft:** George J. Watase, Yukiko M. Yamashita.

**Writing – review & editing:** George J. Watase, Yukiko M. Yamashita.

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
