## [Decision Letter · Decision Letter 0]

16 Feb 2024

Dear Yukiko

Thank you very much for submitting your Research Article entitled 'RNA polymerase II-mediated rDNA transcription mediates rDNA copy number expansion in Drosophila' to PLOS Genetics.

The manuscript was fully evaluated at the editorial level and by four independent peer reviewers. The first three reviewers found the paper highly meritorious, but have suggested numerous edits or small revisions that might improve the text  The 4th review was substantially more strident and highly critical on a very large number of aspects of the paper and argued for a negative decision. We think the strident tone of this reviewer suggests an almost philosophical divide between approaches. Therefore, while we are asking you to consider the review in it's entirety, we are confident you will know which points raise constructive concerns that can be addressed.  Please focus your attentions on those points. 

We therefore ask you to modify the manuscript according to the review recommendations. Your revisions should address as many of the specific points made by each reviewer as is feasible.

1) Provide a mercifully brief summary of the changes you have made for our reference. A point-by-point response is not required.  We are not likely to send the paper bacl to the reviewers.

Yours sincerely,

R. Scott Hawley

Academic Editor

PLOS Genetics

Gregory P. Copenhaver

Editor-in-Chief

PLOS Genetics

Reviewer's Responses to Questions

**Comments to the Authors:**

Reviewer #1: The manuscript “RNA polymerase II-mediated rDNA transcription mediates rDNA copy number expansion in Drosophila” uses imaging of the germ stem cells of the testes to examine how repeat expansion of the rDNA is controlled. rDNA magnification is stimulated in genotypes with reduced numbers of rDNA copies. This group has previously identified a DNA-binding protein Indra that is required for magnification, and further characterize its role in magnification here. Overall this study shows clear data implicating repression of a promoter in the IGS of the rDNA unit by Indra in preventing repeat expansion. The authors link together expression of this IGS promoter, derepression of the R2 retroposon, and induced DNA damage as steps in repeat expansion specifically in the germ stem cells. This is an interesting study, but the significance of observations on nuclear distribution of RNAPII is unclear, and I have a few comments on exactly how these observations may fit together.

Lines 219-231 discuss the localization of RNAPII in the nucleus determined by immunofluorescence imaging. The authors argue that RNAPII is relocalized to rDNA genes at the periphery of the nucleolus, but this is a surprising result that is not clearly laid out in this section. It seems from Figure 5A vs 5B that there is a global relocalization of RNAPII to heterochromatin around the nucleolus, but does this really fit with the amount of repressed rDNA that is in these genotypes? This seems unlikely given how big these regions are and must encompass much more sequence than just rDNA. Could the authors give some sense of how much RNAPII they are detecting from image analysis, perhaps by analyzing the distribution of signal across the nucleus? Further, Figure 5E and 5J categorize nuclei based on interpretations of whether a nucleus has RNAPII at the nucleolus or not, but the authors should give some measure of how much this signal varies between cells.

The authors use alpha-amanitin treatments to show that IGS transcription is RNAPII-dependent. Since they have observed that IGS transcription is correlated with R2 retroposon transcription, they go on to argue that both the IGS are transcribed by RNAPII (lines 233-242). This is surprising, because R2 is thought to be transcribed by RNAPI as part of a pre-rRNA. The model the authors lay out here and in the Discussion explicitly invoke a transcript originating in the IGS, but an alternative is that RNAPII-mediated transcription of the IGS promoter activates the adjacent rDNA RNAPI promoter. The authors might discuss these possibilities in the text or address the existence of long RNAPII transcript originating in the IGS. In many organisms including Drosophila and Arabidopsis short transcripts are produced from IGS regions, and the authors could discuss these results in a broader context.

One caveat is the authors use a very high concentration of alpha-amanitin, and at these concentrations it may also inhibit RNAPIII.

Reviewer #2: In this manuscript the authors provide evidence that the indra protein is a negative regulator of rDNA magnification. Its an interesting study that advances the mechanistic insight into Indra, although some gaps in understanding still remain. I have provided some comments below for the authors’ consideration.

1. In the introduction the authors cite papers that say ERCs cause aging in yeast, but whether they are causal or just correlated has not been resolved-it would be good to acknowledge the controversy. The authors cite the original study but there have been several studies since and the situation may be more complicated, for example see Hotz et al., PNAS, 2022, Zylstra et al., PLoS Biology 2023

2. The asterisks in Figure 1 are tiny and difficult to see.

3. How much is indra overexpressed in the overexpression experiments in Figure 2 and 4? Can the Indra antibody be used to estimate overexpression? If this was done previously, please provide the percent overexpression in the current manuscript with reference.

4. Can the authors use their indra antibody to estimate the depletion in the RNAi condition? If this was done previously, please provide the percent depletion in the current manuscript with reference.

5. Does indra overexpression over generations lead to the eventual loss of rDNA? Would the intermediate state eventually give way to the bobbed state?

6. Can the authors estimate how many R2 containing versus non-containing repeats there are?

7. Indra protein levels are examined. Although they are reduced in the low copy number flies, the amount of change doesn’t seem very substantial. How do the authors propose indra protein levels are regulated? Is the RNA regulated? This question plagued me throughout reading the manuscript. I don’t think the authors need to resolve it, but some speculation about how and why Indra levels are lower when copy number is lower would improve readability.

8. Does Indra interact or colocalize with RNA Pol II?

9. The authors should quantify the effects shown in Figure 7 not just for R2 but for IGS and ETS, to provide more evidence these are RNAP2 transcripts.

10. Figure 6-it looks like RNAP2 is everywhere in the nucleus, and excluded from the nucleolus across all the conditions examined based on the images. How did the authors decide what qualified as signal at the nucleolar periphery.

11. Number of observations should be provided for S2.

Minor-

Probably better to use “containing” than “inserted”

Reviewer #3: In this study, Watase and Yamashita describe how rDNA magnification is regulated in male germline stem cells by zinc finger protein Indira to maintain germline immortality. The Yamashita lab has been steadily uncovering how metazoa regulate rDNA copy number using male germline stem cells from Drosophila as a model system. They previously discovered that rDNA copy number in under the control of Indira and a transposable element R2. R2 upregulation causes double strand breaks (DSBs), which then promotes homologous recombination to increase rDNA copy number. R2 does not have its own promoter but instead its transcription is under control of sequences in the intergenic spacers (IGS) in the rDNA repeats. However, the molecular mechanism of how R2 transcription is regulated was not fully clear.

Here they find that:

1. Low rDNA copy number promotes IGS transcription.

2. Indira represses IGS transcription under WT conditions.

3. Under low copy number levels of Indira decreases.

4. Downregulation of Indira promotes R2 transcription, which then promotes double strand breaks.

5. Indira promotes transcription of R2 via allowing access to pol II.

Conceptual advance: Here they find that the Zinc finger protein Indira that binds to IGS regulates R2 transcription that then promotes increase in copy number by promoting DSBs and recombination. In addition, their data suggests that the parts of the nucleolar compartment are privileged with sparing access to pol II that is under the control of Zinc finger protein Indira.

Overall, they describe a clear set of experiments in this well written manuscript. I have a few comments that should be addressed prior to publication:

1. Figure 1: It would be good to have a schematic of a testis with all the cell types they talk about in the supplementary data so this paper can be read by a broad audience that are the readers of PLOS Gen.

2. Figure 4B – they should mention the genotype. I understand this is an example, but it would be good to know if this what happens in Indira RNAi.

3. Figure 5 – again it would be good to know the genotype of 5A or see pictures of the other genotypes. Is there a pattern to this upregulation? In that do all GSCs in a single testis show an upregulation or is it stochastic under mutant conditions?

4. Figure 6— For me this is a weakest data they have in they have in the manuscript. The idea that Indira regulates RNA pol II levels to regulate IGS is gleaned from Figure 7. Figure 6 shows an almost a global increase in Pol II around the nucleolus. This can be due to many reasons including increased access to IGS sequence. I think this is supplementary data for Figure 7. If they want to include this data, it makes more sense to do a in situ of IGS and show there is an overlap.

5. Figure 7—They should include an Indira RNAi + amanitin condition to bolster their model.

Reviewer #4: This is a review of Manuscript PGENETICS-D-24-00048, “RNA polymerase II-mediated rDNA transcription mediates rDNA copy number expansion in Drosophila,” by Watase and Yamashita.

The manuscript purports to show that the indra gene product binds to rDNA and mediates rDNA magnification in Drosophila. Specifically, the authors make 7 claims:

1 - Indra is a negative regulator of the expression of the Intergenic Spacer in the rDNA

2 - Indra protein level is tightly regulated in Germ Cells

3 - Indra protein regulation is regulated by rDNA multiplicity

4 - IGS expression is thereby controlled by rDNA copy number

5 - RNA Polymerase II is responsible for IGS expression

6 - IGS expression reads through to R2-laden rDNA cistrons

7 - RNA polymerase is responsible for R2 expression

Overall, I find the work described here to be superficial, poorly controlled, and many of the data simply not believable. The central claims are generally unsupported by the data.

Major Concerns

1 - The assays seem to rely on quantitative in situ hybridization, which are not known to be quantitative. The nucleolus and rDNA within it are dynamic in terms of condensation and probe accessibility, as well as copy number. The data in Figure 1E, 2D, etc, even backed up with “best pictures,” are not believable without some form of independent quantification, and some evidence that the FISH signal is sensitive (i.e., zero signal is zero rDNA), proportional (i.e., N% increase in signal is N% increase in rDNA), controlled for bleaching, etc. I am made more skeptical by the images in Figure 2

---

## [Editor Report · Decision Letter 1]

8 May 2024

Dear Yukiko,

Thank you for your very thughtful and detailed revision.  It is truly scholarly/  Based on your revision, we are pleased to inform you that your manuscript entitled "RNA polymerase II-mediated rDNA transcription mediates rDNA copy number expansion in Drosophila" has been editorially accepted for publication in PLOS Genetics. Congratulations!

Yours sincerely,

R. Scott Hawley

Academic Editor

PLOS Genetics

Gregory P. Copenhaver

Section Editor

PLOS Genetics

Comments from the reviewers (if applicable):

**Data Deposition**

http://datadryad.org/submit?journalID=pgenetics&manu=PGENETICS-D-24-00048R1

**Press Queries**

---

## [Editor Report · Acceptance letter]

11 May 2024

PGENETICS-D-24-00048R1 

RNA polymerase II-mediated rDNA transcription mediates rDNA copy number expansion in Drosophila 

Dear Dr Yamashita, 

We are pleased to inform you that your manuscript entitled "RNA polymerase II-mediated rDNA transcription mediates rDNA copy number expansion in Drosophila" has been formally accepted for publication in PLOS Genetics! Your manuscript is now with our production department and you will be notified of the publication date in due course.

With kind regards,

Zsofia Freund

PLOS Genetics

On behalf of:
